# Algorithmic Instabilities
# of Accelerated Gradient Descent

**Amit Attia**
Blavatnik School of Computer Science
Tel Aviv University
amitattia@mail.tau.ac.il

**Tomer Koren**
Blavatnik School of Computer Science
Tel Aviv University, and Google Research
tkoren@tauex.tau.ac.il

## Abstract

We study the algorithmic stability of Nesterov's accelerated gradient method. For convex quadratic objectives, Chen et al. [10] proved that the uniform stability of the method grows quadratically with the number of optimization steps, and conjectured that the same is true for the general convex and smooth case. We disprove this conjecture and show, for two notions of algorithmic stability (including uniform stability), that the stability of Nesterov's accelerated method in fact deteriorates *exponentially fast* with the number of gradient steps. This stands in sharp contrast to the bounds in the quadratic case, but also to known results for non-accelerated gradient methods where stability typically grows linearly with the number of steps.

## 1 Introduction

Algorithmic stability has emerged over the last two decades as a central tool for generalization analysis of learning algorithms. While the classical approach in generalization theory originating in the PAC learning framework appeal to uniform convergence arguments, more recent progress on stochastic convex optimization models, starting with the pioneering work of Bousquet and Elisseeff [6] and Shalev-Shwartz et al. [24], has relied on stability analysis for deriving tight generalization results for convex risk minimizing algorithms.

Perhaps the most common form of algorithmic stability is the so called *uniform stability* [6]. Roughly, the uniform stability of a learning algorithm is the worst-case change in its output model, in terms of its loss on an arbitrary example, when replacing a single sample in the data set used for training. Bousquet and Elisseeff [6] initially used uniform stability to argue about the generalization of empirical risk minimization with strongly convex losses. Shalev-Shwartz et al. [24] revisited this concept and studied the stability effect of regularization on the generalization of convex models. Their bounds were recently improved in a variety of ways [13, 14, 7] and their approach has been influential in a variety of settings (e.g., 18, 16, 9). In fact, to this day, algorithmic stability is essentially the only general approach for obtaining tight (dimension free) generalization bounds for convex optimization algorithms applied to the empirical risk (see 24, 12).

Significant focus has been put recently on studying the stability properties of iterative optimization algorithms. Hardt et al. [17] considered stochastic gradient descent (SGD) and gave the first bounds on its uniform stability for a convex and smooth loss function, that grow linearly with the number of optimization steps. As observed by Feldman and Vondrak [13] and Chen et al. [10], their arguments also apply with minor modifications to full-batch gradient descent (GD). Bassily et al. [5] exhibited a significant gap in stability between the smooth and non-smooth cases, showing that non-smooth GD and SGD are inherently less stable than their smooth counterparts. Even further, algorithmic stability has also been used as an analysis technique in stochastic mini-batched iterative optimization (e.g., 25, 1), and has been proved crucial to the design and analysis of differentially private optimization algorithms [26, 4, 15], both of which focusing primarily on smooth optimization.

35th Conference on Neural Information Processing Systems (NeurIPS 2021).

Having identified smoothness as key to algorithmic stability of iterative optimization methods, the following fundamental question emerges: how stable are *optimal methods* for smooth convex optimization? In particular, what is the algorithmic stability of the celebrated Nesterov accelerated gradient (NAG) method [22]—a cornerstone of optimal methods in convex optimization? Besides being a basic and natural question in its own right, its resolution could have important implications to the design and analysis of optimization algorithms, as well as serve to deepen our understanding of the generalization properties of iterative gradient methods. Chen et al. [10] addressed this question in the case of convex *quadratic* objectives and derived bounds on the uniform stability of NAG that grow quadratically with the number of gradient steps (as opposed to the linear growth known for GD). They conjectured that similar bounds hold true more broadly, but fell short of proving this for general convex and smooth objectives. Our work is aimed at filling this gap.

## 1.1 Our Results

We establish tight algorithmic stability bounds for the Nesterov accelerated gradient method (NAG). We show that, somewhat surprisingly, the uniform stability of NAG grows *exponentially fast* with the number of steps in the general convex and smooth setting. Namely, the uniform stability of $T$-steps NAG with respect to a dataset of $n$ examples is in general $\exp(\Omega(T))/n$, and in particular, after merely $T = O(\log n)$ steps the stability becomes the trivial $\Omega(1)$. This result demonstrates a sharp contrast between the stability of NAG in the quadratic case and in the general convex, and disproves the conjecture of Chen et al. [10] that the uniform stability of NAG in the general convex setting is $O(T^2/n)$, as in the case of a quadratic objective.

Our results in fact apply to a simpler notion of stability—one that is arguably more fundamental in the context of iterative optimization methods—which we term *initialization stability*. The initialization stability of an algorithm $A$ (formally defined in Section 2 below) measures the sensitivity of $A$'s output to an $\epsilon$-perturbation in its initialization point. For this notion, we demonstrate a construction of a smooth and convex objective function such that, for sufficiently small $\epsilon$, the stability of $T$-steps NAG is lower bounded by $\exp(\Omega(T))\epsilon$. Here again, we exhibit a dramatic gap between the quadratic and general convex cases: for quadratic objectives, we show that the initialization stability of NAG is upper bounded by $O(T\epsilon)$.

For completeness, we also prove initialization stability upper bounds in a few relevant convex optimization settings: for GD, we analyze both the smooth and non-smooth cases; for NAG, we give bounds for quadratic objectives as well as for general smooth ones. Table 1 summarizes the stability bounds we establish compared to existing bounds in the literature. Note in particular the remarkable exponential gap between the stability bounds for GD and NAG in the general smooth case, with respect to both stability definitions. Stability lower bounds for NAG are discussed in Sections 3 and 4; initialization stability upper bounds for the various settings and additional uniform stability bounds are detailed in the full version of the paper [3] .

| METHOD | SETTING | INIT. STABILITY | UNIF. STABILITY | REFERENCE |
|--------|---------|-----------------|-----------------|-----------|
| GD | convex, smooth | $\Theta(\epsilon)$ | $\Theta(T/n)$ | Hardt et al. [17] |
| GD | convex, non-smooth | $\Theta(\epsilon + \eta\sqrt{T})$ | $\Theta(\eta\sqrt{T} + \eta T/n)$ | Bassily et al. [5] |
| NAG | convex, quadratic | $O(T\epsilon)$ | $\Theta(T^2/n)$ | Chen et al. [10] |
| NAG | convex, smooth | $\exp(\Theta(T))\epsilon$ | $\exp(\Theta(T))/n$ | (this paper) |

Table 1: Stability bounds introduced in this work (**in bold**) compared to existing bounds. For simplicity, all bounds in the smooth case are for $\eta = \Theta(1/\beta)$. The lower bounds for NAG are presented here in a simplified form and the actual bounds exhibit a fluctuation in the increase of stability; see also Fig. 2 and the precise results in Sections 3 and 4.

Finally, we remark that our focus here is on the general convex (and smooth) case, and we do not provide formal results for the strongly convex case. However, we argue that stability analysis in the latter case is not as compelling as in the general case. Indeed, a strongly convex objective admits a unique minimum, and so NAG will converge to an $\epsilon$-neighborhood of this minimum in $O(\log(1/\epsilon))$ steps from any initialization, at which point its stability becomes $O(\epsilon)$; thus, with strong convexity perturbations in initialization get quickly washed away as the algorithm rapidly converges to the unique optimum. (A similar reasoning also applies to uniform stability with strongly convex losses.)

## 1.2 Overview of Main Ideas and Techniques

We now provide some intuition to our constructions and highlight some of the key ideas leading to our results. We start by revisiting the analysis of the quadratic case which is simpler and better understood.

**Why NAG is stable for quadratics:** Consider a quadratic function $f$ with Hessian matrix $H \succeq 0$. For analyzing the initialization stability of NAG, let us consider two runs of the method initialized at $x_0, \tilde{x}_0$ respectively, and let $(x_t, y_t)$, $(\tilde{x}_t, \tilde{y}_t)$ denote the corresponding NAG iterates at step $t$. Further, let us denote by $\Delta_t^x \triangleq x_t - \tilde{x}_t$ the difference between the two sequences of iterates. Using the update rule of NAG (see Eqs. (1) and (2) below) and the fact that for a quadratic $f$, differences between gradients can be expressed as $\nabla f(x) - \nabla f(x') = H(x - x')$ for any $x, x' \in \mathbb{R}^d$, it is straightforward to show that the distance $\Delta_t^x$ evolves according to

$$\Delta_{t+1}^x = (I - \eta H)\big((1 + \gamma_t)\Delta_t^x - \gamma_t \Delta_{t-1}^x\big).$$

This recursion can be naturally put in matrix form, leading to:

$$\begin{pmatrix} \Delta_{t+1}^x \\ \Delta_t^x \end{pmatrix} = \prod_{k=1}^{t} \begin{pmatrix} (1 + \gamma_k)A & -\gamma_k A \\ I & 0 \end{pmatrix} \begin{pmatrix} \Delta_1^x \\ \Delta_0^x \end{pmatrix},$$

where here $A = I - \eta H$. Thus, for a quadratic $f$, bounding the divergence $\|\Delta_t^x\|$ between the two NAG sequences reduces to controlling the operator norm of the matrix product above, namely

$$\left\| \prod_{k=1}^{t} \begin{pmatrix} (1 + \gamma_k)A & -\gamma_k A \\ I & 0 \end{pmatrix} \right\|.$$

Remarkably, it can be shown that this norm is $O(t)$ for any $0 \preceq A \preceq I$ and any choice of $-1 \leq \gamma_1, \ldots, \gamma_t \leq 1$. (This can be seen by writing the Schur decomposition of the involved matrices, as we show in the full version of the paper [3].[1]) As a consequence, the initialization stability of NAG for a quadratic objective $f$ is shown to grow only linearly with the number of steps $t$.

**What breaks down in the general convex case:** For a general convex (twice-differentiable and smooth) $f$, the Hessian matrix is of course no longer fixed across the execution. Assuming for simplicity the one-dimensional case, similar arguments show that the relevant operator norm is of the form

$$\left\| \prod_{k=1}^{t} \begin{pmatrix} (1 + \gamma_k)A_k & -\gamma_k A_k \\ I & 0 \end{pmatrix} \right\|,$$

where $0 \leq A_1, \ldots, A_t \leq 1$ are related to Hessians of $f$ taken at suitable points along the optimization trajectory. However, if $A_k$ are allowed to vary arbitrarily between steps, the matrix product above might explode exponentially fast, even in the one-dimensional case! Indeed, fix $\gamma_k = 0.9$ for all $k$, and set $A_k = 0$ whenever $k \bmod 3 = 0$ and $A_k = 1$ otherwise; then using simple linear algebra the operator norm of interest can be shown to satisfy

$$\left\| \left( \begin{pmatrix} 0 & 0 \\ 1 & 0 \end{pmatrix} \begin{pmatrix} 1.9 & -0.9 \\ 1 & 0 \end{pmatrix} \begin{pmatrix} 1.9 & -0.9 \\ 1 & 0 \end{pmatrix} \right)^{t/3} \right\| = \left\| \begin{pmatrix} 0 & 0 \\ 2.71 & -1.71 \end{pmatrix}^{t/3} \right\| \geq 1.15^t.$$

**How a hard function should look like:** The exponential blowup we exhibited above hinged on a worst-case sequence $A_1, \ldots, A_t$ that varies significantly between consecutive steps. It remains unclear, however, what does this imply for the actual optimization setup we care about, and whether such a sequence can be realized by Hessians of a convex and smooth function $f$. Our main results essentially answer the latter question on the affirmative and build upon a construction of such a function $f$ that directly imitates such a bad sequence.

Concretely, we generate a hard function inductively based on a running execution of NAG, where in each step we amend the construction with a "gadget" function having a piecewise-constant Hessian (that equals either 0 or the maximal $\beta$); see Fig. 1 for an illustration of this construct. The interval pieces are carefully chosen based on the NAG iterates computed so far in a way that a slightly perturbed

---

[1]Chen et al. [10] give an alternative argument based on Chebyshev polynomials.

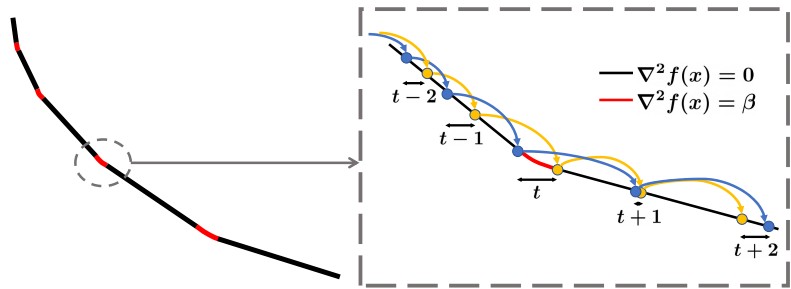

Figure 1: A function (left) constructed of four instantiations of our "gadget" (right) at increasing sizes. During an interval with zero Hessian, the trajectory with the larger momentum gains distance. When reaching an interval with maximal Hessian (depicted here as iteration $t$), the "slow" trajectory experiences a larger gradient which gives it larger momentum and makes it become the "faster" one.

execution would traverse through intervals with an appropriate pattern of Hessians that induces a behaviour similar to the one exhibited by the matrix products above, leading to an exponential blowup in the stability terms. Fig. 2 shows a simulation of the divergence between the two trajectories of NAG on the objective function we construct, illustrating how the divergence fluctuates between positive and negative values, with its absolute value growing exponentially with time. More technical details on this construction can be found in Section 3.

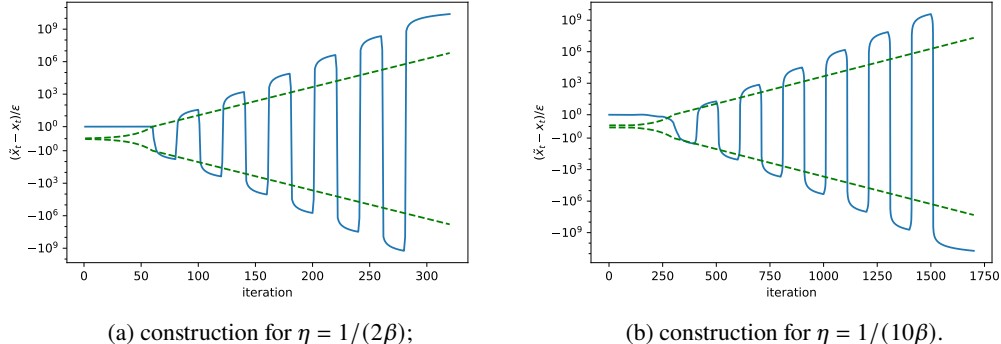

(a) construction for $\eta = 1/(2\beta)$;    (b) construction for $\eta = 1/(10\beta)$.

Figure 2: Divergence between trajectories (in log-scale) along the optimization process for different values of $\eta$. At steps $t = \Theta(i/\eta\beta)$ $(i = 1, 2, \ldots)$ NAG experiences an exponential growth in the divergence, as it reaches an interval with maximal Hessian. The dashed lines depict our theoretical exponential lower bound.

**From initialization stability to uniform stability:** Finally, we employ a simple reduction to translate our results regarding initialization stability to relate to uniform stability in the context of empirical risk optimization, where one uses (full-batch) NAG to minimize the (convex, smooth) empirical risk induced by a sample $S$ of $n$ examples. Concretely, we show that by replacing a single example in $S$, we can arrive at a scenario where after one step of NAG on the original and modified samples the respective iterates are $\epsilon = \Theta(1/n)$ away from each other, whereas in the remaining steps both empirical risks simulate our bad function from before. Thus, we again end up with an exponential increase in the divergence between the two executions, that leads to a similar increase in the algorithmic (uniform) stability: the latter becomes as large as $\Omega(1)$ after merely $T = O(\log n)$ steps of full-batch NAG. The formal details of this reduction can be found in Section 4.

## 1.3 Discussion and Additional Related Work

It is interesting to contrast our results with what is known for the closely related heavy ball method [23]. Classic results show that while for convex quadratic objectives the (properly tuned) heavy ball method attains the accelerated $O(1/T^2)$ convergence rate, for general convex and smooth functions it might even fail to converge at all (see 20). More specifically, it is known that there exists objectives for which heavy ball assumes a cyclic trajectory that never gets close to the optimum; it is then not hard

to turn such a construction to an instability result for heavy ball, as a slight perturbation in the cyclic pattern can be shown to make the method converge to optimum.

Also related to our work is Devolder et al. [11], that analyzed GD and NAG with *inexact first-order information*, namely, in a setting where each gradient update is contaminated with a bounded yet arbitrary perturbation. Interestingly, they showed that in contrast to GD, NAG suffers from an accumulation of errors—which appears analogous to the linear increase in initialization stability the latter experiences in the quadratic case. At the same time, in the general convex case their results might seem to be at odds with ours as we show that even a single perturbation at initialization suffices for extreme instabilities. However, note that they analyze the impact of perturbations on the *convergence rate* of NAG (in terms of objective value), whereas algorithmic stability is concerned with their effect on the actual *iterates*: specifically, initialization stability captures to what extent the iterates of the algorithm might stray away from their original positions as a result of a small perturbation in the initialization point.

Various forms of stability are well-studied in the literature of dynamical systems, with the notion of Lyapunov stability being perhaps the most notable one. While related to our study, Lyapunov stability is concerned with stability of a solution due to a perturbation near a point of equilibrium of the system, and the solution is then said to be stable if the system remains in equilibrium despite this perturbation. In contrast, we are concerned with an arbitrary perturbation to the system (or algorithm) initialization point, which is the case of interest for initialization stability.

Our work leaves a few intriguing open problems for future investigation. Most importantly, it remains unclear whether there exists a different accelerated method (one with the optimal $O(1/T^2)$ rate for smooth and convex objectives) that is also poly($T$)-stable. Bubeck et al. [8] suggested a geometric alternative to NAG that comes to mind, and it could be interesting to check whether this method or a variant thereof is more stable than NAG. Another open question is to resolve the gap between our stability lower and upper bounds for NAG in the regime $\eta \ll 1/\beta$: while our lower bounds have an exponential dependence on $\eta$, the upper bounds do not. Finally, it could be interesting to determine whether the $O(T\epsilon)$ initialization stability bound we have for NAG in the quadratic case is tight (the corresponding uniform stability result is actually tight even for linear losses, but this may not be the case for initialization stability).

## 2 Preliminaries

In this work we are interested in optimization of convex and smooth functions over the $d$-dimensional Euclidean space $\mathbb{R}^d$. A function $f$ is said to be $\beta$-smooth (for $\beta > 0$) if its gradient is $\beta$-Lipschitz, namely, if for all $u, v \in \mathbb{R}^d$ it holds that $\|\nabla f(u) - \nabla f(v)\| \leq \beta\|u - v\|$.

### 2.1 Nesterov Accelerated Gradient Method

The Nesterov Accelerated Gradient (NAG) method [22] we consider in this paper takes the following form. Starting with $x_0$ and $y_0 = x_0$, it iterates for $t = 1, 2, \ldots$:

$$x_t = y_{t-1} - \eta \nabla f(y_{t-1});\tag{1}$$
$$y_t = x_t + \gamma_t(x_t - x_{t-1}),\tag{2}$$

where $\gamma_t = \frac{t-1}{t+2}$ and $\eta > 0$ is a step-size parameter.[2] For a $\beta$-smooth convex objective $f$ and $0 < \eta \leq 1/\beta$, this method exhibits the convergence rate $O(1/\eta T^2)$; for $\eta = 1/\beta$, this gives the optimal convergence rate for the class of $\beta$-smooth convex functions (see [21]). We remark that while NAG appears in several other forms in the literature, many of these are in fact equivalent to the one given in Eqs. (1) and (2). For more details, see the full version of the paper [3].

Throughout, we use the notation $\text{NAG}(f, x_0, t, \eta)$ to refer to the iterates $(x_t, y_t)$ at step $t$ of NAG on $f$ initialized at $x_0$ with step size $\eta$. We sometimes drop the step size argument and use the shorter notation $\text{NAG}(f, x_0, t)$ when $\eta$ is clear from the context.

We will use the following definitions and relations throughout. We introduce the following notation for the momentum term of NAG, for all $t > 0$:

$$m_t \triangleq \gamma_t(x_t - x_{t-1}).\tag{3}$$

---

[2]For concreteness, we focus here on the most common choice of momentum parameters $\gamma_t$, but our results can be adapted to similar settings.

Using this notation, we have that

$$x_t = y_{t-1} - \eta \nabla f(y_{t-1}), \tag{4}$$

$$y_t = x_t + m_t = y_{t-1} - \eta \nabla f(y_{t-1}) + m_t, \tag{5}$$

$$m_t = \gamma_t(m_{t-1} - \eta \nabla f(y_{t-1})). \tag{6}$$

Here, Eq. (6) follows from Eqs. (3) to (5) via

$$m_t = \gamma_t(x_t - x_{t-1}) \qquad\qquad\qquad (\text{Eq. (3)})$$

$$= \gamma_t(y_{t-1} - \eta \nabla f(y_{t-1}) - x_{t-1}) \qquad\qquad (\text{Eq. (4)})$$

$$= \gamma_t(x_{t-1} + m_{t-1} - \eta \nabla f(y_{t-1}) - x_{t-1}) \qquad (\text{Eq. (5)})$$

$$= \gamma_t(m_{t-1} - \eta \nabla f(y_{t-1})).$$

## 2.2 Algorithmic Stability

We consider two forms of algorithmic stability. The first is the well-known *uniform stability* [6], while the second is *initialization stability* which we define here.

**Uniform stability.** Consider the following general setting of supervised learning. There is a sample space $\mathcal{Z}$ of examples and an unknown distribution $\mathcal{D}$ over $\mathcal{Z}$. We receive a training set $S = (z_1, \ldots, z_n)$ of $n$ samples drawn i.i.d. from $\mathcal{D}$. The goal is finding a model $w$ with a small *population risk*:

$$R(w) \triangleq \mathbb{E}_{z \sim \mathcal{D}}[\ell(w; z)],$$

where $\ell(w; z)$ is the loss of the model described by $w$ on an example $z$. However, as we cannot evaluate the population risk directly, learning algorithms will be applied on the *empirical risk* with respect to the sample $S$, given by

$$R_S(w) \triangleq \frac{1}{n} \sum_{i=1}^{n} \ell(w; z_i).$$

In this paper, our algorithm of interest in this context is *full-batch* NAG, namely, NAG applied to the empirical risk $R_S$. We use the following notion of *uniform stability*.[3]

**Definition 1 (uniform stability).** Algorithm $A$ is $\epsilon$-uniformly stable if for all $S, S' \in \mathcal{Z}^n$ such that $S, S'$ differ in at most one example, the corresponding outputs $A(S)$ and $A(S')$ satisfy

$$\sup_{z \in \mathcal{Z}} |\ell(A(S); z) - \ell(A(S'); z)| \le \epsilon.$$

We use $\delta_{A,\ell}^{\text{unif}}(n)$ to denote the infimum over all $\epsilon > 0$ for which this inequality holds.

**Initialization stability.** A second notion of algorithmic stability that we define and discuss in this paper, natural in the context of iterative optimization methods, pertains to the stability of the optimization algorithm with respect to its initialization point. *Initialization stability* measures the sensitivity of the algorithm's output to a small perturbation in its initial point; formally,

**Definition 2 (initialization stability).** Let $A$ be an algorithm that when initialized at a point $x \in \mathbb{R}^d$, produces $A(x) \in \mathbb{R}^d$ as output. Then for $\epsilon > 0$, the initialization stability of $A$ at $x_0 \in \mathbb{R}^d$ is given as

$$\delta_A^{\text{init}}(x_0, \epsilon) = \sup\{\|A(\tilde{x}_0) - A(x_0)\| : \tilde{x}_0 \in \mathbb{R}^d, \|\tilde{x}_0 - x_0\| \le \epsilon\}.$$

## 3 Initialization Stability of NAG

In this section we prove our first main result, regarding the initialization stability of NAG:

**Theorem 3.** *Let $\epsilon, G, \beta > 0$ and $0 < \eta \le 1/\beta$. Consider two initialization points $x_0 = 0, \tilde{x}_0 = \epsilon$. Then, there exists a convex, $\beta$-smooth, $G$-Lipschitz function $f$ that attains a minimum over $\mathbb{R}$, and universal constants $c_1, c_2 > 0$, such that the sequences $(x_t, y_t) = \text{NAG}(f, x_0, t, \eta)$ and $(\tilde{x}_t, \tilde{y}_t) = \text{NAG}(f, \tilde{x}_0, t, \eta)$ satisfy*

$$\delta_{\text{NAG}_t}^{\text{init}}(x_0, \epsilon) \ge |x_t - \tilde{x}_t| \ge \min\left\{\frac{G}{3\beta}, c_2 e^{c_1 \eta \beta t} \epsilon\right\}, \qquad \forall t \in \left\{\lceil \frac{10}{\eta\beta} \rceil (i+2) : i = 1, 2, \ldots\right\}.$$

*Furthermore, for all $t > \lceil \frac{10}{\eta\beta} \rceil \left(\ln \frac{3G}{2\beta\epsilon} + 3\right)$ it holds that $\delta_{\text{NAG}_t}^{\text{init}}(x_0, \epsilon) \ge \frac{G}{3\beta}$.*

---

[3]We give here a definition suitable for deterministic algorithms, which suffices for the context of this paper. Similar definitions exist for randomized algorithms; see for example [17, 13].

In words, the theorem establishes an exponential blowup in the distance between the two trajectories $x_t$ and $\tilde{x}_t$ during the initial $O(1/\epsilon)$ steps, after which the (lower bound on the) distance reaches a constant and stops increasing. Notice that in the blowup phase, an increase in distance happens roughly every $\eta\beta$ steps; indeed, the actual behaviour of NAG on the function we construct exhibit fluctuations in the difference $x_t - \tilde{x}_t$, as illustrated in Fig. 2. We remark that a similar bound holds also for the $y_t$ sequence produced by NAG.

**Construction.** Throughout this section, we will assume without loss of generality that $0 < \epsilon < G/2\beta$. (When $\epsilon \geq G/2\beta$ our result holds simply for a constant function.) To lower bound the initialization stability and prove the theorem, we will rely on the following construction of functions $f_0, f_1, \dots :$ $\mathbb{R} \to \mathbb{R}$. Let the parameters $G, \beta, \eta, \epsilon > 0$ be given, and for all $i \geq 0$ define $n_i \triangleq \lceil 10/\eta\beta \rceil (i+2)$. The construction proceeds as follows:

(i) Let $f_0(x) \triangleq -Gx$;
(ii) For $i \geq 1$:
   - Let $(x_{n_i}, y_{n_i}) = \text{NAG}(f_{i-1}, 0, n_i, \eta)$ and $(\tilde{x}_{n_i}, \tilde{y}_{n_i}) = \text{NAG}(f_{i-1}, \epsilon, n_i, \eta)$;
   - Define $f_i : \mathbb{R} \to \mathbb{R}$ as follows:
$$f_i(x) \triangleq -Gx + \beta \int_{-\infty}^x \int_{-\infty}^y \mathbf{1}\Big[\exists\, j \leq i \text{ s.t. } z \in [y_{n_j}^{\min}, y_{n_j}^{\max}]\Big] dz dy,$$
   where $y_{n_i}^{\min} = \min\{y_{n_i}, \tilde{y}_{n_i}\}$, $y_{n_i}^{\max} = \max\{y_{n_i}, \tilde{y}_{n_i}\}$;
(iii) Let $M = \sup\{i \geq 0 : \max_x \nabla f_j(x) < -\tfrac{1}{2}G, \ \forall\, 0 \leq j \leq i\}$.

Note that the above recursion defines an infinite sequence of functions $f_0, f_1, f_2, \dots : \mathbb{R} \to \mathbb{R}$. Ultimately, we will be interested in the functions $\{f_i\}_{i \leq M}$ which we will analyze in order to prove the instability result. Further, note that $\max_x \nabla f_0(x) < -\tfrac{1}{2}G$, thus $M$ itself is well-defined, possibly $\infty$. The functions constructed above are not lower-bounded (and thus do not admit a minimum). Below we define a lower-bounded adaptation of $f_M$ (assuming $1 \leq M < \infty$, which is proved in Lemma 7 later on). Our modified version of $f_M$, termed $f$ is defined by a quadratic continuation of $f_M$ right of $p \triangleq y_{n_M}^{\max}$, up to a plateau. This construction is defined formally as:

$$f(x) \triangleq \begin{cases} f_M(x) & x \leq p; \\ f_M(p) + \nabla f_M(p)(x-p) + \tfrac{\beta}{2}(x-p)^2 & p < x \leq p - \tfrac{1}{\beta}\nabla f_M(p); \\ f_M(p) - \tfrac{1}{2\beta}\nabla f_M(p)^2 & \text{otherwise.} \end{cases}$$

**Analysis.** We start by stating a few lemmas we will use in the proof of our main theorem. Our focus is on the functions $f_i$ for $0 \leq i \leq M$, deferring the analysis of $f$ to after we establish that $M$ is finite. First, we show that the functions we constructed are indeed convex, smooth and Lipschitz.

**Lemma 4.** *For all $0 \leq i \leq M$, the function $f_i$ is convex, $\beta$-smooth and $G$-Lipschitz.*

**Proof.** The second derivative of $f_i$ is

$$\nabla^2 f_i(x) = \beta \cdot \mathbf{1}\Big[\exists\, j \leq i \text{ s.t. } z \in [y_{n_j}^{\min}, y_{n_j}^{\max}]\Big] \in \{0, \beta\}.$$

Thus, $f_i$ is convex and $\beta$-smooth. We lower bound the first derivative by

$$\nabla f_i(x) = -G + \beta \int_{-\infty}^x \mathbf{1}\Big[\exists\, j \leq i \text{ s.t. } z \in [y_{n_j}^{\min}, y_{n_j}^{\max}]\Big] dz \geq -G,$$

and by the definition of $M$, $\nabla f_i(x) < -G/2$. Hence for all $x \in \mathbb{R}$ we have $\nabla f_i(x) \in [-G, -G/2)$, so $f_i$ is $G$-Lipschitz over $\mathbb{R}$. ∎

Next, we analyze the iterations of NAG on $f_i$ for any given $0 \leq i \leq M$. Fix such index $i$ and consider $(x_t, y_t) = \text{NAG}(f_i, 0, t)$ and $(\tilde{x}_t, \tilde{y}_t) = \text{NAG}(f_i, \epsilon, t)$ for all $t \leq T$ for some $T \geq n_{i+1}$. We introduce the following compact notation for differences between the NAG terms related to the two sequences:

$$\Delta_t^x \triangleq x_t - \tilde{x}_t, \qquad \Delta_t^y \triangleq y_t - \tilde{y}_t, \qquad \Delta_t^f \triangleq \nabla f_i(x_t) - \nabla f_i(\tilde{x}_t), \qquad \Delta_t^m \triangleq m_t - \tilde{m}_t.$$

From the update rules of NAG (Eqs. (4) to (6)), we have that

$$\Delta_t^x = \Delta_{t-1}^y - \eta\Delta_{t-1}^f, \tag{7}$$

$$\Delta_t^y = \Delta_t^x + \Delta_t^m = \Delta_{t-1}^y - \eta\Delta_{t-1}^f + \Delta_t^m, \tag{8}$$

$$\Delta_t^m = \gamma_t(\Delta_{t-1}^m - \eta\Delta_{t-1}^f). \tag{9}$$

Our next lemma below describes the evolution of the differences $\Delta_t^f$ and $\Delta_t^m$ in terms of $\Delta_t^y$.

**Lemma 5.** *For all $t \leq T$,*

$$\Delta_t^f = \begin{cases} \beta\Delta_t^y & \text{if } t \in \{n_j\}_{j=1}^i; \\ 0 & \text{otherwise,} \end{cases} \quad \text{and} \quad \Delta_t^m = \begin{cases} \gamma_t(\Delta_{t-1}^m - \eta\beta\Delta_{t-1}^y) & \text{if } t \in \{n_j+1\}_{j=1}^i; \\ \gamma_t\Delta_{t-1}^m & \text{otherwise.} \end{cases}$$

The following lemma summarise the evolution of the distance between the sequences $y_t$ and $\tilde{y}_t$ at steps $t \in \{n_j\}_{1 \leq j \leq i+1}$ and for $t > n_{i+1}$. The exponential growth is achieved by a balance between the difference in momentum terms and the difference between the sequences.

**Lemma 6.** *For the difference terms $\Delta_t^y$, we have the following:*

(i) *For all $1 \leq j \leq i$, it holds that $\frac{2}{3}\eta\beta|\Delta_{n_j}^y| \leq |\Delta_{n_j+1}^m| \leq \frac{1}{5}\eta\beta|\Delta_{n_{j+1}}^y|$.*

(ii) *For all $0 \leq j \leq i$, it holds that $|\Delta_{n_{j+1}}^y| = y_{n_{j+1}}^{\max} - y_{n_{j+1}}^{\min} \geq 3^j\epsilon$.*

(iii) *For all $t > n_{i+1}$, it holds that $|\Delta_t^y| \geq |\Delta_{n_{i+1}}^y|$.*

Finally, we can show that $f$ is well-defined by proving that $M$ is finite in the following lemma. The bound of $M$ also indicate that after $O(\log\frac{1}{\epsilon})$ steps the two trajectories $y_t, \tilde{y}_t$ reach a constant distance.

**Lemma 7.** *It holds that $1 \leq M \leq \ln\frac{3G}{2\beta\epsilon}$ (in particular, $M$ is finite), and $y_{n_{M+1}}^{\max} - y_{n_{M+1}}^{\min} \geq \frac{G}{3\beta}$.*

Now we can return to our $f$. First, we show it indeed posses the basic properties for Theorem 3.

**Lemma 8.** *The function $f$ is convex, $\beta$-smooth, $G$-Lipschitz and attains a minimum $x^\star \in \arg\min_x f(x)$ s.t. $|x_0 - x^\star| = O\big((G/\eta\beta^2)\log(G/\beta\epsilon)^2\big)$ for $x_0 \in \{0, \epsilon\}$.*

The final lemma we require shows that the distance between the two trajectories is the same for $f$ and $f_M$. This holds true since the two functions coincide for $x \leq p$, after which the iterates reach a plateau which induces similar stability dynamics as the linear part of $f_M$ at $x > p$.

**Lemma 9.** *Let $(x_t, y_t) = \text{NAG}(f, 0, t, \eta)$ and $(\tilde{x}_t, \tilde{y}_t) = \text{NAG}(f, \epsilon, t, \eta)$ be the iterations of $\text{NAG}$ on $f$ from our initialization points. Similarly, for $f_M$, let $(\hat{x}_t, \hat{y}_t) = \text{NAG}(f_M, 0, t, \eta)$ and $(\bar{x}_t, \bar{y}_t) = \text{NAG}(f_M, \epsilon, t, \eta)$. Then for all $t$, we have that $x_t - \tilde{x}_t = \hat{x}_t - \bar{x}_t$ and $y_t - \tilde{y}_t = \hat{y}_t - \bar{y}_t$.*

We defer the proofs of Lemmas 5 to 9 to the full version of the paper [3], and proceed to prove our main result.

**Proof of Theorem 3.** Based on Lemma 9, it suffices to show that the lower bound holds for the function $f_M$. Let $c_1 = \frac{1}{11}\ln(3), c_2 = \frac{4}{5}3^{-3}$. Let $t = n_i$ for some $i \geq 1$. The first case we will deal with is when $i \leq M + 1$. We already established with Lemma 6 that $|\Delta_{n_i}^y| \geq 3^{i-1}\epsilon$. Since $t = n_i = (i+2)\lceil 10/\eta\beta\rceil, i \geq \eta\beta t/11 - 2$. Hence,

$$|\Delta_{n_i}^y| \geq 3^{\eta\beta t/11-3}\epsilon \quad \implies \quad |\Delta_{n_i}^y| \geq \frac{5}{4}c_2 e^{c_1\eta\beta T}\epsilon.$$

To relate to $|x_t - \tilde{x}_t|$,

$$\begin{aligned}
|x_t - \tilde{x}_t| &= |\Delta_{n_i}^x| \\
&\geq |\Delta_{n_i}^y| - |\Delta_{n_i}^m| && \text{(Eq. (8))} \\
&\geq |\Delta_{n_i}^y| - |\Delta_{n_{i-1}+1}^m|\prod_{t=n_{i-1}+2}^{n_i}\gamma_t && \text{(Lemma 5 for } t = n_i, \ldots, n_{i-1}+2) \\
&\geq |\Delta_{n_i}^y| - |\Delta_{n_{i-1}+1}^m| && \text{(Since } t \geq 1 \Rightarrow 0 \leq \gamma_t \leq 1) \\
&\overset{(*)}{\geq} |\Delta_{n_i}^y|\left(1 - \frac{\eta\beta}{5}\right) \\
&\geq \frac{4}{5}|\Delta_{n_i}^y| \geq c_2 e^{c_1\eta\beta T}\epsilon. && (\eta \leq \tfrac{1}{\beta})
\end{aligned}$$

Here, $(*)$ follows from Lemma 6 if $i > 1$ and the case of $i = 1$ follows by combining Lemma 5 and $\Delta_1^m = 0$ which implies that $\Delta_{n_0+1}^m = 0$. If $t > n_{M+1}$ (includes the case of $i > M + 1$ and

$t > \lceil 10/\eta\beta \rceil \left( \ln \frac{3G}{2\beta\epsilon} + 3 \right)$ from Lemma 7), since $t - 1 \geq n_{M+1}$,

$$
\begin{align}
|x_t - \tilde{x}_t| = |\Delta_{t-1}^y - \eta\Delta_{t-1}^f| && \text{(Eq. (7))} \\
= |\Delta_{t-1}^y| && \text{(Lemma 5)} \\
\geq |\Delta_{n_{M+1}}^y|. && \text{(Lemma 6)}
\end{align}
$$

And using Lemma 7 we conclude that $|x_t - \tilde{x}_t| \geq \frac{G}{3\beta}$. Hence, with Lemma 4, $f_M$ holds all properties of Theorem 3 beside attaining a minimum, and using Lemmas 8 and 9, $f$ posses all the properties needed for Theorem 3. ∎

## 4 Uniform Stability of NAG

In this section we present our second main result, regarding the uniform stability of (full-batch) NAG. This is given formally in the following theorem.

**Theorem 10.** *For any $G, \beta, \eta$ and $n \geq 4$ such that $0 < \eta \leq 1/\beta$, there exists a loss function $\ell(w; z)$ that is convex, $\beta$-smooth and $G$-Lipschitz in $w$ (for every $z \in \mathcal{Z}$) and universal constants $c_3, c_4 > 0$, such that the uniform stability of $T$-steps full-batch NAG with step size $\eta$ is*

$$
\delta_{\mathrm{NAG}_T, \ell}^{\mathrm{unif}}(n) \geq \min\left\{ \frac{G^2}{3\beta}, c_4 e^{c_3\eta\beta T} \frac{\beta\eta^2 G^2}{n} \right\}, \qquad \forall\, T \in \left\{ \lceil \frac{10}{\eta\beta} \frac{n}{n-3} \rceil (i+2) \,:\, i = 1, 2, \ldots \right\},
$$

*Furthermore, for all $T \geq \lceil \frac{40}{\eta\beta} \rceil \left( \ln \frac{6G}{\beta\epsilon} + 3 \right)$ it holds that $\delta_{\mathrm{NAG}_T, \ell}^{\mathrm{unif}}(n) \geq \frac{G^2}{3\beta}$.*

The comments following Theorem 3 regarding the exponential blowup and the fluctuating behaviour also apply here. Note also the perhaps surprising inverse dependence on $\beta$ ($\beta\eta^2$ is also $O(1/\beta)$). The dependence can be explained by the fact that smooth optimization over a highly non-smooth yet still $G$-Lipschitz function must have a small step size (with $\eta \leq 1/\beta$) which improves stability.

**Construction.** We denote the given parameters for the theorem with $\hat{G}, \hat{\beta}, \hat{\eta}, n$. We will use the construction from Section 3 with the properties of Theorem 3 in order to create a loss function and samples which will have the same optimization for $t \geq 1$. For the construction we define the following setting of $G, \beta, \eta, \epsilon$:

$$
G = \hat{G}, \qquad \beta = \hat{\beta}, \qquad \eta = \frac{n-3}{n}\hat{\eta}, \qquad \epsilon = \frac{\beta\eta^2 G}{n-3}.
$$

Using these parameters, we obtain $f$ from the construction of Section 3. As we proved in the previous section, this is the function for which Theorem 3 holds. Note that for $T < n_1$, the lower bounds already holds even for quadratics, as we show in the full version of the paper [3]. We also define the following functions,

$$
\ell(w; 1) \triangleq 0, \tag{10}
$$

$$
\ell(w; 2) \triangleq -\beta\eta G w + \beta \int_{-\infty}^{w} \int_{-\infty}^{y} \mathbf{1}[z \in [0, \eta G]] \, dz \, dy, \tag{11}
$$

and further let $\ell(w; 3) = -Gw$, $\ell(w; 4) = Gw$, and $\ell(w; 5) = f(w)$. Note that all are convex, $\beta$-smooth and $G$-Lipschitz. Let $\mathcal{Z} = \{1, 2, 3, 4, 5\}$ be our sample space, we consider the loss function $\ell(w; z)$ over $\mathbb{R} \times \mathcal{Z}$. Our samples of interest are $S = (1, 3, 4, 5, \ldots, 5) \in \mathcal{Z}^n$ and $S' = (2, 3, 4, 5, \ldots, 5) \in \mathcal{Z}^n$. Thus, the empirical risks $R_S, R_{S'}$ corresponding to $S, S'$ are given by

$$
R_S(w) = \frac{1}{n}g_1(w) + \frac{n-3}{n}f(w) = \frac{n-3}{n}f(w), \tag{12}
$$

$$
R_{S'}(w) = \frac{1}{n}g_2(w) + \frac{n-3}{n}f(w). \tag{13}
$$

**Analysis.** The key lemma below shows that we constructed a scenario that reduces the problem of analyzing the uniform stability of full-batch NAG to analyzing its initialization stability on the function $f$ we constructed in Section 3.

**Lemma 11.** *For all $t = 1, 2, \ldots$, we have that*

$$
\mathrm{NAG}(f, 0, t, \eta) = \mathrm{NAG}(R_S, 0, t, \hat{\eta}),
$$

$$
\mathrm{NAG}(f, \epsilon, t, \eta) = \mathrm{NAG}(R_{S'}, 0, t, \hat{\eta}).
$$

Using this key lemma, Theorem 10 follows immediately by setting $c_3 = c_1/4$ and $c_4 = c_2/4$, for the samples $S, S'$ we defined and $z = 3$. As $\ell(w; 3) = -Gw$,

$$|\ell(x_T; 3) - \ell(\tilde{x}_T; 3)| = G|x_T - \tilde{x}_T|,$$

and by using the definitions of $G, \beta, \eta, \epsilon$ we get the two cases of Theorem 3 which lower bound $|x_T - \tilde{x}_T|$ as needed. The full proof of Lemma 11 and Theorem 10 can be found in the full version of the paper [3].

## Acknowledgments and Disclosure of Funding

We thank Naman Agarwal, Yair Carmon and Roi Livni for valuable discussions. This work was partially supported by the Israeli Science Foundation (ISF) grant no. 2549/19, by the Len Blavatnik and the Blavatnik Family foundation, and by the Yandex Initiative in Machine Learning.

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
