# OpenReview forum: "Algorithmic Instabilities of Accelerated Gradient Descent"
_NeurIPS.cc/2021/Conference — NeurIPS 2021 Poster_

### Official Review · Reviewer_tbGy · 2021-07-13

**Rating:** 7
**Confidence:** 2

**Summary:**

This paper studies the stability of Nesterov accelerated gradient in the initialization and in the sense of uniform stability. The authors prove lower bounds on the stability parameters, that diverge exponentially in the number of iterations, by building clever one-dimensional adversarial examples. This proves that in the general convex smooth case, Nesterov accelerated gradient can be a lot more instable than in the quadratic case.

**Limitations And Societal Impact:**

In general, the authors brought a clean answer to a restricted theoretical question, but I would like to see a discussion on the potential insights that could be gained from this. See the main review for suggestions.

**Main Review:**

The core of this paper is a technical construction of an adversarial example where Nesterov accelerated gradient is instable. I could not check all the details but I appreciated the good exposition and I believe the math to be clean.

The counter-example built by the authors does answer rigorously to the question of uniform stability as it is asked by the authors; however the example seems artificial and very adversarial. As explained in the introduction, the concept of uniform stability was originally used to study the generalization of empirical risk minimization; one could wonder whether the conclusions drawn in this paper transpose to more realistic learning settings. For instance, what happens when the functions in the finite sum are no longer adversarial but i.i.d. according to some law? This perspective might be far-reaching, but I would appreciate to have the opinion of the authors on the subject.

The counter-example of Section 3 is in dimension 1. Could there be a result in larger dimensions? I am asking because the authors use the mean-value theorem in page 14, applied to the gradient of the function. This mean-value theorem does not hold for multi-variate functions.


Minor comments:
- In Theorem 3, why do the authors study G-Lipschitz functions? Could we expect the result to hold without the requirement for the function to be Lipschitz, and without the minimum in the lower-bound?
- l.204: Did the authors mean O(log 1/\varespilon) ?

**Time Spent Reviewing:**

3,5

---

> ### Author Response · Authors · 2021-08-10
> **Response to review**
>
> Thank you for your thoughtful review!  We first respond to the main points in your review; specific replies follow below.
>
> > “the example seems artificial and very adversarial… For instance, what happens when the functions in the finite sum are no longer adversarial but i.i.d. according to some law?”
>
> This is a very good point - indeed, our main focus was on uniform (worst case) stability, which is the most prevalent notion in the ML literature for analyzing generalization of algorithms.  We currently do not know how to extend our results to weaker forms of “average stability” that consider an expectation over the training sample - we will give this some more thought before the final version.  Thanks for this comment!
>
>
> > “Could there be a result in larger dimensions?”
>
> Our results are of course true in higher dimensions (by embedding the 1D function in $\mathbb R^d$) and already show an exponential growth in stability in any dimension.  So we actually view it as a strength of our result: it applies already in d=1!  It might be possible to slightly strengthen the result by exploiting more dimensions; we currently do not know whether and how to do so.
>
> ## Specific replies to minor comments:
>
>
> 1. Recall that we are claiming a lower bound: that is, even if the function is Lipschitz, and even if it is lower bounded, the result holds.  In other words, you could remove the claims about Lipschitzness and boundedness and Theorem 3 would still hold true, but its statement would become a bit weaker.  (In particular, for the construction, you could choose the Lipschitz constant G arbitrarily and make the second term in the minimum arbitrarily large.)
>
> 2. You are correct, we will fix this typo in the final version.  Thanks!

---

> > ### Comment · Reviewer_tbGy · 2021-08-15
> > **Thanks**
> >
> > I thank the authors for their clarifications. I also have taken a look at the other reviews. I wish to keep my grade and my low confidence score.

---

### Official Review · Reviewer_TrR8 · 2021-07-14

**Rating:** 6
**Confidence:** 1

**Summary:**

This paper presents two kinds of Instabilities of Accelerated Gradient Descent. Although with many novel results at the first glance, almost of them  are confusing.

**Limitations And Societal Impact:**

Yes

**Main Review:**

Strengths: 1. This paper proposes the notion of initialization stability and proves that the NAG is initialization unstable. To this end,  the authors prove many technical lemmas.

2. The authors proved the uniform instability of full batch NAG. Many technical lemmas are proved.


Weakness: This paper needs many efforts to improve.

1. The overclaim in the abstract and introduction.  They say they disprove this conjecture proposed in [ Chen et al. 2018]. However, in Sec. 4, they proved the full batch version!  No any stochasticity of the algorithm and the so-called uniform instability actually comes from the initialization stability proposed by the authors. They do not prove the real uniform instability.

2. The initialization stability is similar to the notion of stability in dynamics and PDE. This is reasonable because we may not know whether the dynamic is convergent or not.  However,  in the optimization community,   NAG can be easily guaranteed to converge. And then, two points must get very closed after enough iterations. The initialization stability is useless for any convergent iterates.  The authors can prove the initialization instability because under the conditions given in Theorem 3 one sequence is convergent and the other one is divergent.

3. The uniform stability is proposed by literature and widely used because it can be employed to generalization errors in training problems.  Nevertheless, the authors do not tell us why to introduce the initialization stability and how it is used. It seems the initialization stability is only used for the uniform instability of NAG.


**Time Spent Reviewing:**

6

---

> ### Author Response · Authors · 2021-08-10
> **Response to review**
>
> Thanks for the feedback—please see below our responses to the main points you raised.
>
> > “they proved the full batch version!  No any stochasticity of the algorithm...”; “They do not prove the real uniform instability”
>
> Indeed, we focus on full-batch NAG -- **Nesterov’s canonical algorithm is a non-stochastic full-batch algorithm!**  (Recall that fast accelerated rates are truly meaningful only in exact, non-stochastic optimization.)  Naturally, the conjecture of Chen et al. 2018 also actually concerns full-batch NAG.  **Uniform stability is perfectly relevant for deterministic algorithms**---in fact, the early work of Bousquet & Elisseeff (JMLR, 2002) introduced this concept for analyzing regularized ERM, a purely deterministic procedure.
>
>
> > “NAG can be easily guaranteed to converge. And then, two points must get very closed after enough iterations”
>
> Fundamentally, **this is not true in the general (non-strongly) convex case**: there could be many minimizers to which the optimizer might converge to, and so stability in parameter space is not guaranteed even if the algorithm is convergent.  While this might seem like a degenerate case, it is actually a highly relevant case for ML and a common barrier for establishing generalization bounds (especially for unregularized algorithms)!
>
>
> > “It seems the initialization stability is only used for the uniform instability of NAG”
>
> This is correct: you may view initialization stability as a technical tool (to argue about uniform stability) rather than a key concept in its own right.  That said, we would still argue that it is a very simple, natural and easy to interpret notion that distills a main source of instability in gradient methods (and our formal reduction to uniform stability highlights this connection).

---

> > ### Comment · Reviewer_TrR8 · 2021-08-12
> > **I may misunderstand something.**
> >
> > I have read the rebuttal. And I think I may have made something wrong. The stability is much different from the uniform stability introduced by [Hardt M, Recht B, Singer Y. Train faster, generalize better: Stability of stochastic gradient descent[C]//International Conference on Machine Learning. PMLR, 2016: 1225-1234]. Considering my concerns have been addressed, I raised the score. But due to that I failed to understand the paper, I decreased my confidence.

---

### Official Review · Reviewer_boQt · 2021-07-14

**Rating:** 5
**Confidence:** 4

**Summary:**

It is very interesting and attractive to investigate the algorithmic instabilities for the famous Nesterov accelerated gradient descent method. This concepts  are very related to ordinary differential equation, such as initialization stability corresponding to continuous dependence of solution about initialization for long time. The uniform stability is for the objective function. The authors tries to use the so-called uniform stability to describe the NAG under noise cannot obtain the desired result.

The new concept --- uniform stability is a novel point.

**Limitations And Societal Impact:**

The content  for the new concept uniform stability and comparison with initialization stability is not enough.

**Main Review:**

This paper introduce a new concept --- uniform stability to describe the algorithms' stability, which is an originality. But the quality and clarity is not enough. The authors have not shown the clear intuition for why the new concept --- uniform stability is important, just do the computation. Moreover, it looks very interesting that the authors do deeper investigation. If they can compare the stability concept with the approximating ODEs,

Su et al. (2016)  A Differential Equation for Modeling Nesterov's Accelerated Gradient Method: Theory and Insights
                             Journal of Machine Learning Research, 17(153), 1–43, 2016.

Shi et al. (2021) Understanding the Acceleration Phenomenon via High-Resolution Differential Equations
                             Mathematical Programming, Series A,  https://doi.org/10.1007/s10107-021-01681-8

this paper will become more attractive.

If the author do more effort and deeper investigation, I would like to improve the my review result.

**Time Spent Reviewing:**

35

---

> ### Author Response · Authors · 2021-08-10
> **Response to review**
>
> Thanks for the feedback—please see below our responses to the main points you raised.
>
> > “The authors have not shown the clear intuition for why the new concept --- uniform stability is important, just do the computation”
>
> Uniform stability is **by no means a new concept**: it dates back at least to Bousquet & Elisseeff (JMLR, 2002) and constitutes a well established tool for analyzing generalization of learning algorithms. Please see the 2nd paragraph of our introduction where we discuss the relevant literature and the significance of uniform stability (and algorithmic stability more generally).
>
>
> > “If they can compare the stability concept with the approximating ODEs … this paper will become more attractive”
>
> On the surface, stability of ODEs with respect to their initial conditions is indeed relevant to our study.  That said, there are several crucial differences: (1) the behaviour of continuous time processes is very different from that of discrete-time optimization algorithms and the discretization procedure is key, as have been observed in prior work; (2) stability bounds (both initialization and uniform) tend to depend strongly on the step size parameter (e.g., see Table 1 in the paper) which is absent from the continuous-time approximations.  In any case, this is an interesting point of view and we will add a discussion relating to the literature on ODEs approximations (thank you for the references, we will discuss them as well).

---

### Official Review · Reviewer_Vh9U · 2021-07-15

**Rating:** 9
**Confidence:** 4

**Summary:**

This paper disproves Chen et al.'s conjecture that Nesterov accelerations leads a quadratic growth of the uniform stability, by showing for adequate notions of stability that NAG detoriates exponentially fast.

**Limitations And Societal Impact:**

Y.

**Main Review:**

This is an excellent paper. The topic is important: stability of acceleration of the gradient descent. The presentation is really clear, in particular I found section 1.2 "overview of main ideas and techniques" insightful. It is rare to have a pedagogical presentation in the last iteration of NeurIPS.

Despite these praises, I have some minor concerns that may, or not, be addressed in a rebuttal.
- There is no citation associated to "initialization stability" that is a $\ell_\infty$ control that is quite common in sensitivity analysis. The authors apply it to algorithms in this paper, but at the end, it is the same concept.
- Only NAG is studied here, but one may ask if the analysis is really linked to the specific value of $t-1/t-2$ or if more general momentum or inertial methods could be studied with the same analysis.
- The paper can be seen as a worst case analysis, with the construction of "gadgets", I would like to know if we may recover Chen et al's results for a wider class of functions: what about polynomials? convex-semialgebraic functions?
- There is a lot of interesting results in appendix D and E not discussed in the main text, I guess the authors could squeeze a bit some derivation to be able to present at least what is contained in them.

Note: the paper has 31 pages of supplementary materials, mostly proofs of lemmas. The community should at some point recognize that no one is able to carefully check 7 theoretical papers of this length in the span of a month. I believe the proof strategy is correct -- I obviously, and unfortunately, didn't check the hundreds of equations included.

**Time Spent Reviewing:**

2

---

> ### Author Response · Authors · 2021-08-10
> **Response to review**
>
> Thank you for your insightful review and for the high praise!
>
>
> > “There is no citation associated to "initialization stability" that is a $\ell_\infty$ control that is quite common in sensitivity analysis”
>
> Initialization stability is indeed related to stability notions common in control theory and this warrants some discussion---we will include in the final version.  Note however that in control one is usually concerned with slightly different flavors of stability: e.g., Lyapunov stability pertains to stability of systems w.r.t. perturbations in their state but “at steady state”, i.e., near to a point of equilibrium; other common sensitivity notions study robustness of dynamical systems with respect to misspecifications of the model parameters.  We will discuss those in the revision, and we will be happy to hear (in your final review) of other stability concepts in control that we might have missed.
>
>
> > “one may ask if the analysis is really linked to the specific value of $t-1/t-2$ or if more general momentum or inertia methods could be studied with the same analysis”
>
> Other momentum settings which have similar accumulations (as in Claims 15 and 20) should yield the same results, but other variations in the algorithm might lead to very different behavior. For example, the heavy ball method does not even obtain acceleration in the general case of smooth function (and may even not converge, depending on the tuning). On the other hand, Section C in the supplementary analyzes few other (equivalent) variants of NAG for which our lower bound does hold. This is a very good point---we will comment more in the final version.
>
>
> > “I would like to know if we may recover Chen et al's results for a wider class of functions”
>
> Outside the quadratic case, having a non-constant Hessian makes arguing about upper bounding the stability of NAG challenging. Finding such a wider class of natural functions could be an interesting topic for future work.
>
>
> > “There is a lot of interesting results in appendix D and E not discussed in the main text“
>
> Indeed, the supplementary includes some more interesting results that we did not have room for---we will try to squeeze in the relevant statements in the main text of the final version.

---

> > ### Comment · Reviewer_Vh9U · 2021-08-19
> > **After answers / other reviews**
> >
> > Thanks for your answers. I wish to keep my rating at 9, since my concerns were minors, and I believe the authors can address them in the final version as stated by them. I also looked the other reviews, and I don't share most of their concerns (novelty, full-batch, NAG easy convergence proof, etc).

---

### Decision · Program_Chairs · 2021-09-27

**Decision:**

Accept (Poster)

**Comment:**

The paper studies uniform stability of Nesterov accelerated gradient descent for smooth convex optimization and proves that, unlike in the quadratic case and contrary to a conjecture by [Chen et al., 2018], the error can accumulate exponentially fast (as opposed to quadratically fast, which happens for quadratics). The crux of the approach is in introducing clever constructions of one-dimensional adversarial examples. Overall, the exposition of the paper is clear and the paper appears technically sound (as far as it was possible to verify by reviewers). The paper adds a solid contribution to understanding of stability of accelerated/momentum methods.